# Effect of Neonatal Hearing Screening Results on the Lost to Follow-Up at the Diagnostic Level

**DOI:** 10.3390/healthcare11121770

**Published:** 2023-06-15

**Authors:** Grażyna Greczka, Piotr Dąbrowski, Monika Zych, Witold Szyfter

**Affiliations:** Department of Otolaryngology and Head and Neck Surgery, University of Medical Sciences, 60-355 Poznań, Poland

**Keywords:** hearing screening, risk factors, hearing loss, newborns

## Abstract

(1) Background: An important part of any neonatal hearing screening program is monitoring diagnostic visits to confirm or exclude the presence of hearing loss. In addition, time plays an important role in the diagnosis. We identified the number of children who came for a diagnostic visit and analyzed the time of the first audiological visit, depending on the result of the hearing screening test performed in the first days of a child’s life and the presence or absence of risk factors of hearing impairment. (2) Methods: We analyzed 6,580,524 children, of which 8.9% required further diagnostics. The mean time of follow-up diagnostic visit in the analyzed group was 130 days and differed due to the presence or absence of risk factors for hearing loss before and after the neonatal period. (3) Results: Although the risk of hearing loss in children with risk factors is 2.31 to 6.38 times higher than in children without risk factors depending on the result of the screening test, more than 40% of parents do not report to scheduled audiological visits. (4) Conclusions: Doctors, nurses, and midwives who screen hearing at the neonatological level play an important role in educating parents about the possibility of hearing loss in a child and the need for an audiological examination.

## 1. Introduction

There are many models of newborn hearing screening programs [1] in the world. An important element of them is to encourage parents to go with their children to any follow-up. In addition, time plays an important role in the diagnosis of hearing loss. Early detection of hearing defects and early rehabilitation allow for the proper development of speech and child’s intellect. Detection of hearing loss in 3–6 months of age, together with appropriate prostheses and rehabilitation, gives a good chance of speech development. In the case of children with hearing loss acquired later in life, early prosthesis and auditory training allow speech that has developed physiologically to be preserved (2).

Thanks to the Polish Universal Neonatal Hearing Screening Program (PUNHSP), every newborn child in Poland has a chance of early detection of hearing impairment. The first examination is performed on the second or third day after delivery. In the event of risk factors (RFs) for hearing loss, abnormal results of the otoemission test (OAE), or no test, the child is referred for further diagnostics, which, according to the program assumptions, should start before the age of 3 months [2]. Between 2003 and 2020, the mean annual number of children tested using OAE was 360,965, and 32,353 (8.9%) children were referred to a higher diagnostic level. According to information from the central PUNHSP database, only half of them reported for further diagnostics [3]. This phenomenon is not the same in different groups of children and depends on the presence of risk factors and the result of the hearing screening test. Many countries that conduct auditory screening face the problem of reporting for further diagnostic tests to detect hearing defects, which is why it is so important to gather experience on how to deal with this issue [4,5,6]. In Poland, the PUNHSP program is supervised by a non-governmental organization; only medical procedures are financed by the national health fund. The aim of this article is to show the impact of the hearing screening result and the presence of RFs for hearing loss on the parent’s appearance at the follow-up visit for full audiological diagnostics.

## 2. Materials and Methods

In the analysis, 6,580,524 children born from January 2003 to October 2020 were considered. The data were divided according to the results of the screening test and the presence of risk factors of hearing loss in the neonatal (familial hearing loss, craniofacial anomalies, complex congenital anomalies, premature birth (gestational age < 34 weeks), ototoxicity, TORCH infections, low birth weight, Apgar < 4 in 1st min, Apgar < 6 in 5th min, hyperbilirubinemia, bacterial meningitis, intensive care > 7 days, and respirator support (mechanical ventilation)) and post-neonatal period (suspicion of hearing loss and/or delayed speech development, serious infections, syndrome of congenital defects associated with hearing loss, degenerative diseases of the nervous system or sensory/motor neuropathy, head damage, recurrent or prolonged otitis media with effusion, ototoxic medications). The percentage of children reporting for further diagnostics and the percentage of children diagnosed with hearing loss in particular groups were analyzed in detail. Hearing screening results with the otoacoustic emission method were divided as follows: doubtful result in both ears 0.3%, doubtful result in one ear and correct in the other ear 0.4%, doubtful result in one ear and control (abnormal OAE result or child did not pass OAE testing criteria) in the other ear 0.1%, correct in both ears 94.7%, correct in one ear and control in the other ear 1.7%, control in both ears 1.0%, and no test in both ears 1.9%. The screening results were divided into three parts, correct if the OAE criteria were met (pass), control if the OAE criteria were not met (refer), and doubtful if the result indicated neither correct nor control. In the further part of the analysis, the results of doubtful and control OAE were treated as requiring further diagnostics as control (Figure 1). Next, the results were divided into four groups depending on the presence or absence of risk factors (RFs) for the neonatal period and after the neonatal period: 1. no RFs during the neonatal period and no RFs after the neonatal period, 2. RFs during the neonatal period and no RFs after the neonatal period, 3. RFs during the neonatal period and no RFs after the neonatal period, and 4. RFs during the neonatal period and RFs after the neonatal period (Table 1). Statistical analysis (Statistica 13.0 and R version 4.2.1) software were used to perform statistical analyses. Descriptive statistics such as median, minimum, and maximum, as well as lower and upper quartile, were calculated for variables of the continuous type. Percentages are given for categorical variables. The Kruskal–Wallis ANOVA test was used to find the difference in the time of follow-up diagnostic visit in individual groups. The probability of developing hearing loss in individual groups was given in the form of odds ratios. All calculations were performed at the significance level α = 0.05. Our database is built on a relational database engine. The application Server had Spring framework running RESTful API.

## 3. Results

The mean time of follow-up diagnostic visit in the analyzed group was 130 days and differed due to the presence or absence of RFs for hearing loss before and after the neonatal period—Kruskal–Wallis test: H (3, *N* = 427,919) = 3402.098; *p* = 0.0001 (Table 2). No RFs for the neonatal period and post-neonatal period were observed in 6,303,746 children (95.79%), no RFs for the neonatal period and RFs for the post-neonatal period in 6310 (0.10%), RFs for the neonatal period and no RFs for the post-neonatal period in 264,351 (4.02%), and RFs for the neonatal period and RFs after the neonatal period in 6117 (0.09%). In the study population, 582,862 children were identified as needing to report for further diagnosis to confirm or exclude hearing impairment at the next stage of the PUNHSP program. Audiological diagnosis was performed in 368,217 children, including 89,801 children with correct results of the OAE test without any RFs, which theoretically did not require diagnosis according to the assumptions of the PUNHSP program. A diagnostic visit was registered in 47.77% of children requiring further diagnosis.

### 3.1. No RFs during the Neonatal Period and No RFs after the Neonatal Period

In this group (n = 6,303,746), parents of children whose screening test showed the correct results of the OAE test in one ear and doubtful results in the other ear reported the least frequently; as many as 94.1% of them did not report for control tests, and the percentage of children diagnosed with hearing loss in this group was 9.8% (Table 1). Among children with doubtful results in both ears, 58.2% of parents did not report for further diagnosis, and 2.7% of children were diagnosed with hearing loss. In children with doubtful results in one ear and control in the other ear, 48.1% of parents did not report for further diagnostics, and hearing loss was diagnosed in 6.9% of children. Among children with correct results, who theoretically did not have to report to the diagnostic level, hearing loss was detected in 1.8% of the diagnosed cases. In children with normal results in one ear and control in the other, 51.6% of parents did not report for further diagnostics, and hearing loss was diagnosed in 4.6% of diagnosed children. Among children with results of control in both ears, 51.0% of parents did not report to further diagnostics, and hearing loss was diagnosed in 15.6% of children. Among children with no hearing screening, 71.6% of parents did not report for further diagnosis, and hearing loss was diagnosed in 3.0% of children.

### 3.2. RFs of the Neonatal Period and No RFs after the Neonatal Period

There were 264,351 children in the group with at least one risk factor for neonatal hearing loss and without RFs for the post-neonatal period. In the case of children with doubtful results in both ears, 54.5% of parents did not report for further diagnostics, and hearing loss was diagnosed in 12.9% of children. This means that RFs increase the risk of hearing loss in this group of children by more than five-fold (OR = 5.3). Among the children with doubtful results in one ear and correct results in the other, 46.3% of their parents did not report for further diagnostics, and 23.8% were diagnosed with hearing loss. The chance of developing hearing loss in this group increased three-fold (OR = 2.87) in the event of a risk factor. Among children who had a doubtful result in one ear and control in the other, 47.9% of parents did not report for further diagnosis, and 32.2% of children were diagnosed with hearing loss, which means that there is a chance of hearing loss in this group of children; in the case of the presence of a risk factor, it increases by six times (OR = 6.38). In the case of children with normal screening results, 42.6% of parents did not report for further diagnosis, and hearing loss was detected in 1.3% of children (OR = 0.02). In the case of children with correct results for the screening test in one ear and control in the other, 40.5% of children did not report for further diagnostics, and hearing loss was detected in 11.5% of diagnosed children, which means that the chance of hearing loss with concomitant RFs increases almost three-fold in this group of children (OR = 2.66). Among children with abnormal results in both ears, 45.4% of parents did not report for further diagnosis, and 32.9% of children were diagnosed with hearing loss. This means that the chance of hearing loss in the event of CR in this group of children increases almost threefold (OR = 2.65). Among children who were not screened at the first level, 45.8% did not report for further diagnostics, and 6.6% of children were diagnosed with hearing loss. This means that the chance of having hearing loss doubled when there were RFs for hearing loss (OR = 2.31) (Figure 2).

### 3.3. No RFs during the Neonatal Period and RFs after the Neonatal Period

In the group of children with no RFs for neonatal hearing loss and no RFs for post-neonatal hearing loss, 6310 children were registered. In this group of children, the lack of reporting at the diagnostic level ranged from 15.8% in children without screening to 38.7% in children with correct results of the OAE test in one ear and doubtful results in the other. Hearing loss was diagnosed in 32.3% of children with doubtful results in both ears, 27.7% of children with doubtful results in one ear and correct results in the other ear, 28.9% of children with doubtful results in one ear and control results in the other ear, 6.4% of children with correct results in both ears, 20.1% of children with correct results in one ear and control results in the other ear, 45.7% of children with correct results in both ears, and 15.7% of children without hearing screening.

### 3.4. RFs during the Neonatal Period and RFs after the Neonatal Period

In the group of children with RFs for neonatal hearing loss and RFs for post-neonatal hearing loss, 6117 children were registered. In this group of children, the lack of reporting at the diagnostic level ranged from 3.6% in children with doubtful results in one ear and control in the other ear to 19.0% in children with doubtful results in both ears. Hearing loss was diagnosed in 34.4% of children with doubtful results in both ears, 48.8% of children with doubtful results in one ear and correct results in the other, 33.3% of children with doubtful results in one ear and control in the other ear, 4.5% of children with correct results in both ears, 20.7% of children with correct results in one ear and control in the other ear, 49.6% of children with correct results in both ears, and 20.2% of children without screening.

## 4. Discussion

Conducting a follow-up visit within an appropriate time for infants with suspected hearing loss detected during the first stage of screening is very important for the proper development of the child. Many countries around the world struggle with the problem of encouraging parents to report their children for detailed audiological diagnostics in a timely manner. The literature rarely describes the time of reporting depending on the result of the hearing screening test in the first days of a child’s life and the presence of RFs for hearing damage. Various countries in Europe report the average time to definitive diagnosis for children with hearing screening as ranging from 2 weeks in Malta to 12 months in Ukraine, while this ranged from 1 month in Cyprus to 55 months in Moldova for children without hearing screening [1]. In the United States, it is estimated that approximately 35% of infants who are not screened for hearing loss do not receive the recommended audiological evaluations necessary to diagnose hearing loss by 3 months of age. The authors had no information as to whether these infants had been examined by an audiologist or whether the results had been entered into the database of the hearing screening program. Such children are referred to as “Lost to follow-up” [4,7,8]. Of the 27 countries where the analysis of reporting to the diagnostic level was performed, almost half had more than 30% of the children going without diagnosis, where 18 out of 41 (44%) observations were lost to follow up. As a result, nearly half of hearing screening programs lose too many children who do not attend thorough audiological assessment [1,9]. The lack of regional or national databases on hearing screening tests, as well as often irregular data collection, affects the quality of many screening programs. This is due to the lack of child-tracking programs requiring audiological diagnosis and treatment [10]. There is a statistically significant difference in the time to present at the diagnostic level of neonatal hearing screening due to the outcome of the hearing screening test and the presence of risk factors for hearing impairment. This is due to the fact that parents of children with burdens such as congenital defect syndromes first save their children’s health and lives, leaving the diagnosis of their child’s hearing defects for a later time. It is important to make these parents aware of the need for audiological diagnostics, because it affects the later development of the child’s speech and intellect. In the future, creation of a mobile application for the parents of children with an incorrect screening test result or risk factors for hearing damage could enable the parents to be notified of the need to perform audiological diagnostics.

In the PUNHSP, data on reporting at the diagnostic level concern only information received from centers participating in the program and registering visits to the CDB. The last detailed research conducted in Poland in 2014 shows that not 47.6% but 83.6% of children present for hearing screening [3]. In the case of children registered with PUNHSP in Poland without concomitant RFs for hearing loss, the percentage of children without hearing screening who did not attend the diagnostic visit reached 71.6%, and in the case of children with a correct result in one ear and a doubtful result in the other ear, this reached 94.1%. In the case of children with RFs for the neonatal period, children with doubtful results in one ear and correct results in the other were the least likely to undergo diagnostic tests. In this case, 54.5% of children remained without a diagnosis. In other cases, depending on the result of the OAE test, the lack of diagnosis was between 40.5% and 46.5%. In the group of children without RFs for the neonatal period and with RFs after the neonatal period, the lack of diagnoses ranged from 15.8% to 35.4%, while in the group of children with RFs for both the neonatal period and after the neonatal period, no final diagnoses were made. Results ranged between 9.0% and 19.0% depending on the results of the OAE test at the diagnostic level. However, there is no current information on how many parents report to audiological diagnostics privately.

In the literature review, it can be seen that deficiencies in parental education were associated with a high rate of lost to follow-up. In such cases, appropriate data management strategies were used to avoid such situations [10]. A US study [11] identified four areas where there were barriers to the continuation of audiological diagnosis, namely the lack of service system capacity, lack of knowledge of service providers, barriers for families to access services, and information gaps. The authors identified areas where programs could be improved in the future: improving data systems, providing medical support, linking to centers outside the program, supporting parents, and raising awareness of the importance of early detection of hearing loss. It was emphasized that an important aspect is better communication between healthcare providers and the patient monitoring system. In some German Länder, centers have centralized databases to identify and monitor children who have failed a valid test result or have not actually been screened [12]. These experiences show the need for further improvement in the standardization, collection, and reporting of information in PUNHSP. Incomplete reporting of all confirmatory test results to the PUNHSP program, including ongoing diagnostic assessments and monitoring, results in the loss of valuable information needed to help children who have not yet received the recommended diagnosis and rehabilitation. Hence, it increases the chance of developmental delays. The continuous implementation of best practices and helping families to understand the importance of identifying hearing loss early is essential to the future success of deaf or hard-of-hearing children and early intervention programs [5].

Another aspect of this analysis is the ability to undergo hearing screening but still have significant hearing loss. There is very little literature on the likelihood of a false positive OAE. In PUNHSP, hearing loss was recorded in 1.8% of children who had correct screening test results and no noted RFs for hearing loss. In this case, parents usually present themselves to the diagnostic level due to poor language development or behavioral problems. A hearing screening procedure may fail for several reasons: a false positive screening result, retrocochlear pathology with intact hair cell function, or progressive/delayed hearing loss [5].

The strength of this study is the large number of children in the analyzed group and the uniformity of the program. All centers work the same way and perform the same tests during diagnosis. A limitation of the study is that we do not know how many parents were screened at private facilities.

## 5. Conclusions

Doctors, nurses, and midwives who perform hearing screening at the neonatological level play an important role in educating parents about the possibility of hearing loss in a child and the need for an audiological examination. Therefore, funds for the supervision of reporting at the diagnostic level of PUNHSP are limited. Increasing the funds in the PUNHSP for the improvement of the quality of the system controlling the cases of lost to follow-up to control visits would allow for a decisive tightening of the PUNHSP system.

## Figures and Tables

**Figure 1 healthcare-11-01770-f001:**
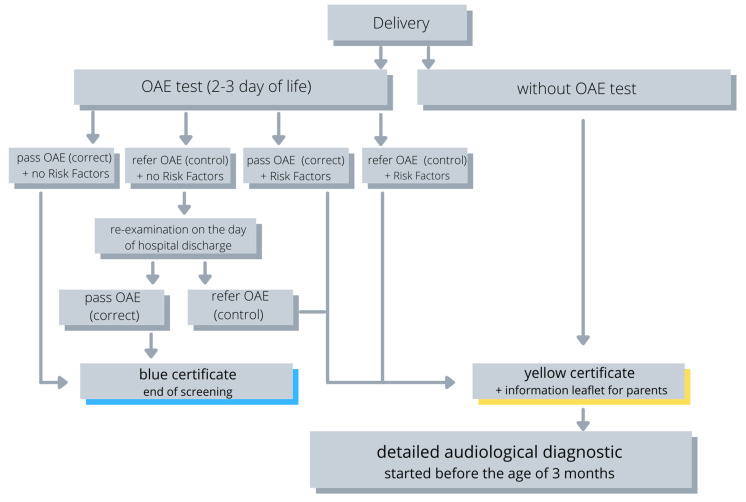
Procedure for hearing screening at PUNHSP.

**Figure 2 healthcare-11-01770-f002:**
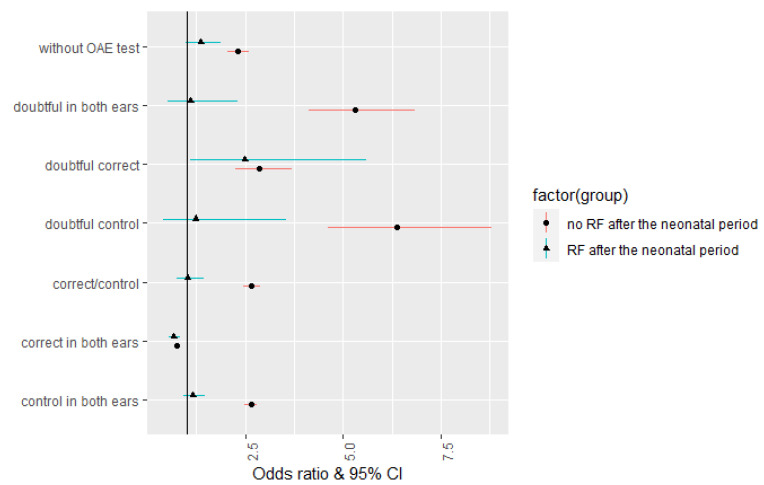
Risk of hearing loss by hearing screening result (RFs of hearing loss).

**Table 1 healthcare-11-01770-t001:** The results of audiological diagnosis and the number of children lost to follow-up depending on the result of the hearing screening performed at the neonatal ward.

No RFs during the Neonatal Period and No RFs after the Neonatal Period							
	Normal hearing	Hearing loss	% of hearing loss	Diagnosis	Without diagnosis	% without diagnosis	Total
Doubtful in both ears	5936	165	2.7%	6101	8500	58.2%	14,601
Doubtful correct	1177	128	9.8%	1305	20,702	94.1%	22,007
Doubtful control	1743	130	6.9%	1873	1737	48.1%	3610
Correct in both ears	89,801	1649	1.8%	91,450	5,906,212	98.5%	5,997,662
Correct Control	45,398	2208	4.6%	47,606	50,800	51.6%	98,406
Control in both ears	22,490	4160	15.6%	26,650	27,719	51.0%	54,369
Without OAE test	31,184	957	3.0%	32,141	80,950	71.6%	113,091
No RFs during the neonatal period and RFs after the neonatal period							
	Normal hearing	Hearing loss	% of hearing loss	Diagnosis	Without diagnosis	% without diagnosis	Total
Doubtful in both ears	42	20	32.3%	62	34	35.4%	96
Doubtful correct	47	18	27.7%	65	41	38.7%	106
Doubtful control	27	11	28.9%	38	9	19.1%	47
Correct in both ears	2496	170	6.4%	2666	987	27.0%	3653
Correct Control	525	132	20.1%	657	200	23.3%	857
Control in both ears	294	247	45.7%	541	161	22.9%	702
Without OAE test	603	112	15.7%	715	134	15.8%	849
RFs during the neonatal period and no RFs after the neonatal period							
	Normal hearing	Hearing loss	% of hearing loss	Diagnosis	Without diagnosis	% without diagnosis	Total
Doubtful in both ears	745	110	12.9%	855	1024	54.5%	1879
Doubtful correct	544	170	23.8%	714	615	46.3%	1329
Doubtful control	166	79	32.2%	245	225	47.9%	470
Correct in both ears	127,703	1724	1.3%	129,427	96,251	42.6%	225,678
Correct Control	6032	781	11.5%	6813	4632	40.5%	11,445
Control in both ears	4692	2299	32.9%	6991	5803	45.4%	12,794
Without OAE test	5443	386	6.6%	5829	4927	45.8%	10,756
RFs during the neonatal period and RFs after the neonatal period							
	Normal hearing	Hearing loss	% of hearing loss	Diagnosis	Without diagnosis	% without diagnosis	Total
Doubtful in both ears	42	22	34.4%	64	15	19.0%	79
Doubtful correct	22	21	48.8%	43	5	10.4%	48
Doubtful control	18	9	33.3%	27	1	3.6%	28
Correct in both ears	3884	181	4.5%	4065	487	10.7%	4552
Correct Control	264	69	20.7%	333	42	11.2%	375
Control in both ears	282	277	49.6%	559	56	9.1%	615
Without OAE test	305	77	20.2%	382	38	9.0%	420
No RFs after the neonatal period		Risk of hearing loss no RFs during the neonatal period vs. RFs during the neonatal period					
		OR		95 % CI:		*p*-value	
Doubtful in both ears		5.31		4.12 to 6.84		*p* < 0.0001	
Doubtful/correct		2.87		2.24 to 3.69		*p* < 0.0001	
Doubtful/control		6.38		4.63 to 8.80		*p* < 0.0001	
Correct in both ears		0.74		0.69 to 0.79		*p* < 0.0001	
Correct/control		2.66		2.44 to 2.90		*p* < 0.0001	
Control in both ears		2.65		2.49 to 2.81		*p* < 0.0001	
Without OAE test		2.31		2.05 to 2.61		*p* < 0.0001	
RFs after the neonatal period		Risk of hearing loss no RFs during the neonatal period vs. RFs during the neonatal period					
		OR		95 % CI:		*p*-value	
Doubtful in both ears		1.1		0.52 to 2.31		*p* = 0.8011	
Doubtful/correct		2.49		1.11 to 5.59		*p* = 0.0267	
Doubtful/control		1.23		0.42 to 3.56		*p* = 0.7059	
Correct in both ears		0.68		0.55 to 0.85		*p* = 0.0006	
Correct/control		1.04		0.75 to 1.44		*p* = 0.8161	
Control in both ears		1.17		0.92 to 1.48		*p* = 0.1959	
Without OAE test		1.36		0.99 to 1.87		*p* = 0.0611	

**Table 2 healthcare-11-01770-t002:** The child’s age (in days) at the time of the first audiological visit.

Descriptive Statistics	*N*	Minimum	Lower Quartile	Median	Upper Quartile	Maximum
No RFs during the neonatal period and no RFs after the neonatal period	207,126	5.0	51.0	80.0	118.0	6389.0 (17.5 years)
RFs during the neonatal period and no RFs after the neonatal period	150,874	4.0	60.0	88.0	123.0	5471.0 (15.0 years)
RFs during the neonatal period and RFs after the neonatal period	5473	3.0	61.0	90.0	133.0	3435.0 (9.4 years)
No RF during the neonatal period and RFs after the neonatal period	4744	38.0	67.0	101.0	199.0	4906.0 (13.4 years)
Total	368,217	3.0	55.0	84.0	121.0	6389.0 (17.5 years)

## Data Availability

If someone wants to get the original data, please contact the author.

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
