# Peer review of "Effect of Neonatal Hearing Screening Results on the Lost to Follow-Up at the Diagnostic Level"

_healthcare, 2023, doi:10.3390/healthcare11121770_

Round 1

Reviewer 1 Report

This article examines the follow-up rates and incidence of true hearing loss by classifying the patients who required retesting during newborn hearing screening into four risk patterns in Poland.

There are some problems with the discussion as described below. We would like to revisit the paper with corrections to the following issues

The discussion only reiterates the results of the previous report on "lost to follow up" in screening and the present results, but does not discuss why there is a difference in follow up and diagnosis rates among the four groups. Is there any meaning in the classification into these four groups? I could not read that from this paper. Also, what measures can be taken to prevent "lost to follow up" based on the results of this risk classification?

What does Control mean in terms of OAE results, which are not well described in other literature on PUNHSP? What is the meaning of "Control" in the OAE results?

I don't understand how to look at Figure 1, it says RF after neonatal period, but how about with or without RF in the neonatal period? Please describe the Figure legend. However, if the Odds ration in Table 2 is used as it is in the graph, this graph may not be necessary.

Author Response

Dear Reviewer, 

Thanks for the valuable remarks, we have changed the discussion and hope it is now more accessible.

Best regards,

Authors

Reviewer 2 Report

Effect of neonatal hearing screening results on the lost to fol-2 low up at the diagnostic level (Healthcare 2023, 11, x FOR PEER REVIEW)

Greczka and co-workers have studied the impact of hearing screening result and the presence of risk factors for hearing loss on the parent's appearance at the follow-up visit for full audiological diagnostics in a large polish cohort or more than 6 million newborns. 

1. General comments: 

This is study reports detailed data in a large nationwide cohort of newborns. The issue is very important due to the critical time window with respect to final diagnosis and therapy. However, the manuscript is cumbersome to read and needs major revision. 

2. Major comments.

Introduction.

Page 1, line 27: `Also, time plays an important role in the diagnosis of hearing loss.´ Please add 1-2 sentences to reason why time is so critical for final diagnosis and subsequent treatment. 

Material and Methods.

Please provide detailed information on the source of the data, i.e. data acquisition, central database, software for database.

Please define risk factors in and after the neonatal period in detail.

Please provide a flow diagram of the `PUNHSP programme´ particular with the role of the parents. This would be extremely helpful to the reader.

Statistics: Please report descriptive statistics with respect to distribution.

Results.

Table 1 is extremely cumbersome to read, please edit!

Table 2: Please report only median, quartiles and range because the data are not normally distributed due to extreme outliers with late testing. Please provide the number of newborns in the left column. 

Please re-organize this chapter: 1) Characteristics of the newborns and detailed risk factors 2) Time to diagnosis of hearing loss, and 3) time to diagnosis of hearing loss with respect to risk factors.

Discussion.

The authors started the Discussion with a large paragraph on time to diagnosis of hearing loss and proportion of lost to follow up. However, they do not integrate their own results! Therefore, please reorganize this chapter also by re-stating the aim of your study and answering the questions posed in the introduction. In this context, please organize the Discussion from the specific to the general: your findings to the

literature, to theory, to practice. Begin by re-stating the hypothesis tested in this analysis and answering the questions posed in the introduction.

Please discuss your own results in context of the literature rather than giving an overview on the literature. 

Please compare the prevalence of conspicuous results in the hearing screening and in final diagnoses of hearing loss in your programme with data of the literature in detail.

Finally, please provide a paragraph on strengths and limitation of this analysis. 

Conclusions.

`In Poland, the PUNHSP programme is supervised by a non-governmental organisation; only medical procedures are financed by the national health fund.´  This information belongs in the Introduction or should be earlier placed in the Discussion.

Author Response

Dear Reviever,

thank you for this valuable remarks. In reference to the above comments, I am sending a response to the reviewer's suggestions.

Best regards,

Authors

Round 2

Reviewer 1 Report

The content has been improved and is more readable.I thought the discussion incorporated what the authors had learned from the results.

The following things need to be understood a little better.

I can’t still understand what ”control" mean. What the difference between doubtful and control. 

I thought that being classified as doubtful is when the OAE results in a refer.

However, the explanation on page 2, line 65 states that a refer would be placed in control.In that case, what is the status of doubtful?

In the figure on page 3, there are only pass and refer results, so if pass is correct and refer is doubtful, what is control?

Minor

On page 2, from the end of line 70 to line 72, different fonts are used.

The figure on page 3 does not have a title. I think it needs to be mentioned.

Author Response

please see the attachment, thank you!
